# Robust support vector machine based on sample screening

Junnan Guo
*School of Mathematics and Statistics*
*Chongqing Jiaotong University*
Chongqing, China
622220150038@mails.cqjtu.edu.cn

Weikai Li
*School of Mathematics and Statistics*
*Chongqing Jiaotong University*
Chongqing, China
leeweikai@outlook.com

Jin Hu*
*School of Mathematics and Statistics*
*Chongqing Jiaotong University*
Chongqing, China
jhu@cqjtu.edu.cn

*Abstract*—Support Vector Machine (SVM) is a prevalent classifier within machine learning, yet its robustness is compromised by the presence of contaminated samples. Such samples, often encountered in practical scenarios, represent deviations from expected data distribution and can include irrelevant or adversarial instances. To enhance SVM's resilience, Fuzzy SVM (FSVM) was introduced, leveraging sample weights to mitigate the impact of outliers. However, FSVM has been criticized for its tendency to sacrifice accuracy, leading to inconsistent performance gains. To address this issue, we introduce a novel robust SVM framework designed to counteract the effects of adversarial samples during training. Our approach involves dynamically setting the weights of samples with substantial loss values to zero, thereby diminishing the influence of outliers. It can be viewed as incorporating sample screening during training process, thus decreasing training time. This modification is particularly effective in scenarios where training data may be tainted with labels that are intentionally misleading. The experimental findings demonstrate that this strategy significantly enhances classifier's robustness against contaminated data, without compromising accuracy. This robust SVM presents a promising solution for improving the reliability of SVMs in real-world applications, where data integrity can be a critical concern.

*Index Terms*—Support Vector Machine, Robust Support Vector Machine, Fuzzy Support Vector Machine, Sample Screening

## I. INTRODUCTION

Support Vector Machine (SVM) [1]–[3] is a machine learning algorithm widely used in pattern recognition, data mining, and statistical learning. Its basic principle involves finding support vectors within a dataset to construct a hyperplane that effectively separates different categories. SVM exhibits excellent performance in handling high-dimensional data and nonlinear problems, making it a powerful and versatile classifier.

Significant achievements in SVM have been made in various domains, such as disease diagnosis [4], [5], mechanical fault detection [6], [7], human behavior detection [8], and text classification [9]. Although SVM has been successfully applied in various domains, its standard loss function is still unbounded, making it sensitive to outliers and lacking sufficient robustness. Robustness is crucial for handling noise and outliers for real-world applications and adapting to different data distributions. In machine learning, robustness refers to a model's resistance to perturbations or noise in input data. A robust model maintains stable and reliable performance in the face of various challenges, including noise, missing values, or outliers. Robustness also helps prevent a model from being overly sensitive to attacks or malicious interference, particularly in security-sensitive domains such as finance, healthcare, and safety.

To enhance the robustness of SVM, researchers have proposed the following approaches:

1) Robustness can be enhanced by adjusting the weights of training samples. Fuzzy SVM [10] reassigned sample weights based on their distance to the separating hyperplane, making the model less sensitive to outliers by reducing their impact.

2) Robustness can be enhanced by enhancing the loss function. Truncated hinge loss function [11] and rescaled hinge loss function [12] set an upper limit on the loss value to limit the influence of samples with large losses on training. Other loss functions such as Pinball hinge loss function [13], [14] SCAD (smoothly clipped absolute deviation) loss function [15] and truncated Huber loss function [16] are also used to improve robustness.

3) Robustness can be enhanced by performing sample selection or feature selection. Selecting training samples during the training process [17]–[20] reduces the influence of outliers on the model but also shrinks the training scale, which is conducive to reducing training time and cost.

4) Robustness can be enhanced by combining SVM with other machine-learning approaches. Combination of SVM and ensemble learning [21]–[23] constructs a more powerful and generalizable overall model by combining multiple SVM models, reduces the risk of overfitting of individual models, and improves robustness of the overall model. Leveraging deep learning for feature extraction leads to more accurate and reliable results, especially in complex tasks like brain tumor classification

The work described in this paper was supported by the National Natural Science Foundation of China under Grant 62176032, 62276034, 62306051 and 62481540175, Joint Training Base Construction Project for Graduate Students in Chongqing under Grant JDLHPYJD2021016, Group Building Scientific Innovation Project for universities in Chongqing under Grant CXQT21021. * Corresponding author.

[5]. Combining multiview learning (MVL) with SVM tackles the challenges of handling data from multiple perspectives or feature sets and enhances classifier's ability to perform robustly across different views [24].

Inspired by the truncated hinge loss function, our approach introduces a novel sample weight iteration strategy to mitigate the influence of outliers on the SVM model. This innovative approach transforms the training process of SVM into a dynamic sample screening procedure. In this new SVM (sample screening SVM), we begin by pretraining a standard SVM, which is the foundation for subsequent sample weighting. We then introduce a threshold parameter, $\delta$, determining the cutoff point for sample inclusion. This adaptive sample selection process is a robust mechanism for outlier removal, enhancing the model's resilience to external interference. Moreover, reducing the number of samples considered during the training process not only improves robustness of the SVM model but also yields significant computational advantages. In summary, our proposed SVM effectively balances outlier removal with model robustness, paving the way for enhanced performance in real-world applications.

This paper makes following notable contributions to enhance robustness of SVM:

1) Novel sample weight strategy: We propose a unique sample weight iteration method that effectively reduces the impact of outliers on the SVM model. Unlike traditional approaches, our approach reduces the weights of some target samples to 0, which improves the accuracy of the model.
2) Integration of sample screening in SVM training: By incorporating sample screening directly into the SVM training process, our approach improves robustness and significantly reduces computational costs. This approach prunes non-essential support vectors, which decreases the overall training time without sacrificing performance accuracy.
3) Empirical validation across multiple domains: We conduct extensive experiments across various datasets to demonstrate versatility and effectiveness of our approach. Results show superior robustness and computational efficiency compared to existing fuzzy SVM approaches.

The remainder of this paper is organized as follows: In Section 2, we briefly introduce FSVM and the sample screening approach. In Section 3 we propose a novel RSVM-SS algorithm and discuss its relationship with FSVM and other approaches. Section 4 conducts numerical analysis on specific datasets. Finally, in Section 5, we summarize and conclude our findings.

## II. RELATED WORKS

### A. FSVM: Fuzzy support vector machine

Let's consider a training set $T$ with $N$ training samples, which is represented as $T = \{(x_1, y_1), (x_2, y_2), \cdots, (x_N, y_N)\}$, where $x_i \in \Re^d$,

$y_i \in \{1, -1\}, i = 1, 2, \ldots, N$. For a model $f$, the predicted value $\widehat{y}_i$ of sample $\boldsymbol{x}_i$ is given by $\widehat{y}_i = \text{sgn}(f(\boldsymbol{x}_i))$. The solution of FSVM [25] involves first training a standard SVM, then assigning corresponding sample weights $\widetilde{w}_i$ based on the function values $f(\boldsymbol{x}_i)$ of samples $\boldsymbol{x}_i$. At this point, the original problem and the dual problem of SVM take the following forms:

Original Problem:

$$\min_{\boldsymbol{w}, b} \frac{1}{2} \|\boldsymbol{w}\|_2^2 + C \sum_{i=1}^{N} \widetilde{w}_i l(f(\boldsymbol{x}_i)) \qquad (1)$$

Dual Problem:

$$\min_{\boldsymbol{\alpha}} \quad \frac{1}{2} \sum_{i=1}^{N} \sum_{j=1}^{N} \alpha_i \alpha_j y_i y_j \langle \boldsymbol{x}_i, \boldsymbol{x}_j \rangle - \sum_{i=1}^{N} \alpha_i$$
$$\text{s.t.} \quad \sum_{i=1}^{N} \alpha_i y_i = 0, \quad i = 1, \cdots, N \qquad (2)$$
$$0 \leq \alpha_i \leq C \cdot \widetilde{w}_i, i = 1, \cdots, N$$

where $C$ is the regularization parameter, $l(\cdot)$ represents the loss function, which usually uses the hinge loss function ($l_{\text{hinge}}(u) = \max\{0, 1 - u\}$), $\alpha_i$ represents the Lagrange coefficient corresponding to sample $\boldsymbol{x}_i$, and $\langle \cdot, \cdot \rangle$ denotes the inner product.

In the case of linear SVM, $f(\boldsymbol{x}) = \boldsymbol{w}^T \boldsymbol{x} + b$. For nonlinear SVM problems, the inner product $\langle \cdot, \cdot \rangle$ is replaced by the kernel function $K(\cdot, \cdot)$. A common kernel function is the Radial Basis Function (RBF), also known as the Gaussian kernel, defined as:

$$K(\boldsymbol{x}_i, \boldsymbol{x}_j) = \exp\left(-\gamma \|\boldsymbol{x}_i - \boldsymbol{x}_j\|^2\right) \qquad (3)$$

where $\gamma$ is a parameter used to control the width of the kernel function.

Various types of sample weights are proposed [25], [26]:

$$\widetilde{w}_i = 1 - \frac{|f(\boldsymbol{x}_i)|}{\max(|f(\boldsymbol{x}_i)|) + \Delta} \qquad (4)$$

$$\widetilde{w}_i = \frac{2}{1 + \exp(\mu|f(\boldsymbol{x}_i)|)} \qquad (5)$$

$$\widetilde{w}_i = \frac{1}{1 + |f(\boldsymbol{x}_i)|} \qquad (6)$$

where $\Delta$ is a tiny constant to prevent division by zero, and $\mu$ is a predefined parameter used to control the decay rate.

In [12], RSVM-RHHQ (robust support vector machine using rescaled hinge loss function and half-quadratic) modifies the loss function ($l_{\text{hinge}}(u) = \max\{0, 1-u\}$) to a bounded loss function ($l_{\text{rhinge}}(u) = \beta[1 - \exp(-\eta l_{\text{hinge}}(u))]$). According to [12], the solution of RSVM-RHHQ is equivalent to an iterative FSVM. The iterative sample weights correspond to:

$$\widetilde{w}_i^{(s+1)} = \exp\left(-\eta l\left(f^{(s)}(\boldsymbol{x}_i)\right)\right) \qquad (7)$$

where $s$ represents the iteration step, $f^{(s)}(\boldsymbol{x}_i)$ is the function value of sample $\boldsymbol{x}_i$ at the $s$-th iteration, and $\eta$ is a predefined constant.

### B. Sample screening

Based on the SVM formulation, its prediction values depend only on the final support vectors, which typically constitute a small subset of all samples. Sample screening [17] is an approach derived from this insight, aiming to filter out non-support vectors during the training process, thereby reducing training time and enhancing the model's efficiency and accuracy. It proposed a safe sample screening method that estimates the contribution of each sample to the optimal hyperplane to exclude those unlikely to become support vectors. [18] further extended this idea by introducing a robust SVM sample screening technique to handle datasets containing noise and outliers. Common goal of these methods is to reduce the size of the training set, lower computational complexity, and maintain or even enhance the model's performance.

### III. RSVM-SS: ROBUST SUPPORT VECTOR MACHINE BASED ON SAMPLE SCREENING

In standard SVM, sample weights can be considered fixed to 1. In FSVM, the model pretrains a standard SVM as the basis for subsequent sample weighting. (4), (5), and (6) assign smaller sample weights to points with larger values of $|f(\boldsymbol{x}_i)|$, which implies that these points are farther from the separating hyperplane. Since the loss value of points correctly classified outside the support vectors is 0, motivated by FSVM, we propose RSM-SS (Robust support vector machine based on sample screening), which reduces the sample weights to 0 for certain samples after training the standard SVM. These weights are defined as follows:

$$\widetilde{w}_i = \begin{cases} 0, & y_i f(\boldsymbol{x}_i) < \delta \\ 1, & \text{otherwise} \end{cases} \tag{8}$$

where $\delta$ is a predefined threshold, and typically, $\delta$ takes a negative value. The smaller the absolute value of $\delta$, the more sample weights of misclassified points are assigned as 0. If $\delta$ is set to 0, all sample weights of misclassified points are set to 0.

When substituting (8) into (1), the resulting optimization problem resembles the self-paced learning (SPL) approach. SPL [27], [28] begins training with a small subset of samples, progressively relaxing the selection criteria to include all samples eventually. The main goal of SPL is to help the model escape local optima and achieve better overall performance. In contrast, our RSVM-SS algorithm focuses on training with screened samples, aiming to reduce the impact of outliers on the model. It can be considered a sample selection to remove potential outliers. Unlike the approaches in [17], [18], which aim to filter out non-support vectors as much as possible, our approach retains correctly classified non-support vectors. Although this increases training time, it experimentally results in higher generalization performance and better accuracy on the test set.

Like FSVM, our RSVM-SS algorithm consists of the following steps:

Step 1 : Initialization. Set all sample weights $\widetilde{w}_i$ to 1. Train a standard SVM and compute $f(\boldsymbol{x}_i)$ for each sample.

Step 2 : Iteration. Update $\widetilde{w}_i$ based (8). Then train another SVM.

Step 3 : Stop when the stop condition is met or the maximum number of iterations is reached.

### IV. EXPERIMENTS

In this section, we conduct experiments for our RSVM-SS model on 7 datasets. Table I shows the basic information of these datasets.

TABLE I
DETAILS OF BENCHMARK DATASETS

| Dataset | Samples | Positive | Negative | Features |
|---|---|---|---|---|
| Wdbc | 569 | 357 | 212 | 30 |
| Transfusion | 748 | 178 | 570 | 4 |
| Pima Indians Diabetes | 768 | 268 | 500 | 8 |
| A5a | 6414 | 1569 | 4845 | 122 |
| Musk | 6598 | 1017 | 5581 | 166 |
| Svmguide1 | 3089 | 2000 | 1089 | 4 |
| Magic Gamma | 19020 | 12332 | 6688 | 10 |

We conducted experiments with various $\delta$ values, including -0.2, -0.5, -1, and -2. Through our experimentation, we determined that a threshold value of -1 proved to be particularly effective. Since the features were normalized before the experiment, setting $\delta$ to -1 resulted in an optimal number of screened samples. This balance ensured that the benefits of improving robustness and reducing computation time were not compromised. Moreover, this threshold selection mitigated concerns about the model's generalization, as it struck a suitable balance between outlier removal and preserving the representativeness of the dataset. Overall, this approach not only enhances robustness of the model but also improves computational efficiency.

In practical experiments, we found that when $\delta$ is set to a relatively large value (e.g., 0), it may lead to all remaining samples belonging to the same class, rendering training infeasible. One solution is to set the weight adjustment to a very small value instead of 0 when performing sample screening, and then continue training. Another approach is to make $\delta$ not fixed. For instance, we can sequentially calculate the violation degree of misclassified samples and set the $q$-th quantile as the threshold $\delta_q$. By doing so, samples with smaller violation degrees are preserved, avoiding excessive filtering.

For comparison with other robust SVMs, we employ the following SVM models:
- Standard SVM
- RSVM-RHHQ: Sample weights use (7). Referring to [12], when interference samples are relatively few (0 % and 10 %), $\eta$ is set to 0.5 ; while interference samples are more abundant (20%), $\eta$ is set to 2 . The optimization method adopts the iterative FSVM method in (7) with two iterations.
- FSVM-7: Sample weights use (4), referring to [26].
- FSVM-8: Sample weights use (5), referring to [26], with $\mu$ set to 0.7.
- FSVM-9: Sample weights use (6), referring to [25].
- Our RSVM-SS: Sample weights use (8). For a fixed screening threshold, $\delta$ is set to -1, and the corresponding SVM

model is named RSVM-SS-1 (denoted as RS-1 in Fig. 1 to 15). For the quantile screening method, quantile is set to 0.5, and the corresponding SVM model is named RSVM-SS-q0.5 (denoted as RS-q0.5 in Fig. 1 to 15).

Other parameters are set as follows: $C$ in SVM is set to 1. When using the Gaussian kernel, $\gamma$ is calculated as $\gamma = 1/\left(d \cdot \sigma_X^2\right)$ , where $d$ is the dimensionality of the feature space, and $\sigma_X^2$ is the feature variance of the training samples. The experiments utilize 10-fold cross-validation. For each dataset, we sequentially take 0 %, 10 %, and 20 % of the samples with opposite labels as interference data. Accuracy of models on the test set is used as a metric for evaluating their robustness. All datasets are standardized before training.

### A. Model robustness

First, we compare the accuracy of different models when interference samples are present in the dataset. In Fig. 1 to 7, we will compare the accuracy of each model (mean%). Each dataset corresponds to six scenarios, which are combinations of two types of kernels and three levels of noise, namely, (linear, 0%), (linear, 10%), (linear, 20%), (rbf, 0%), (rbf, 10%), and (rbf, 20%).

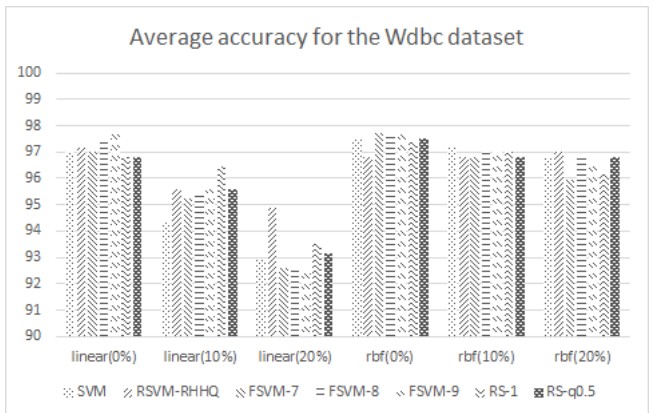

Fig. 1. Average accuracy of the Wdbc dataset. The horizontal axis represents the combinations of different noise levels and kernel function types, where "linear" represents the linear kernel, "rbf" represents the radial basis function kernel, and the numbers represent the proportion of interference samples. The vertical axis represents the average accuracy of the models.Fig. 2 to 7 take the same horizontal and vertical axes.

By analyzing results obtained from our experiments, it is evident that our proposed model significantly enhances the accuracy of SVM in the presence of interference samples. This improvement is particularly noteworthy as it effectively counters the adverse effects of increased noise levels in the dataset. Notably, our model consistently outperforms the traditional SVM model across various datasets, indicating its robustness and efficacy in handling challenging scenarios.

Upon thorough examination, it's evident that our model consistently demonstrates superior performance, manifesting the highest accuracy improvement across five of the datasets analyzed. However, RSVM-RHHQ displays the highest accuracy improvement in the remaining two datasets; nevertheless,

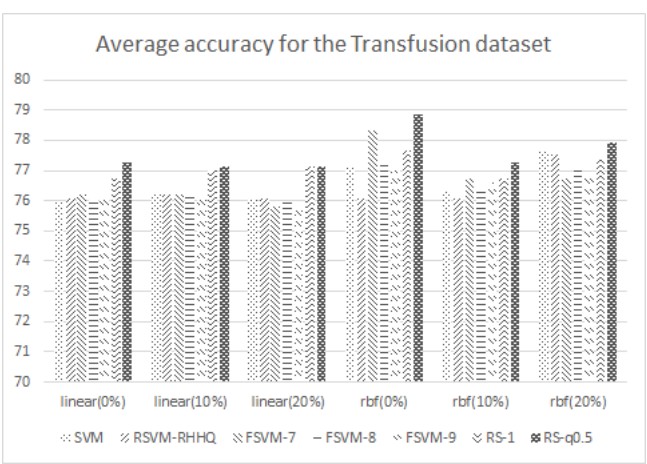

Fig. 2. Average accuracy for the Transfusion dataset .

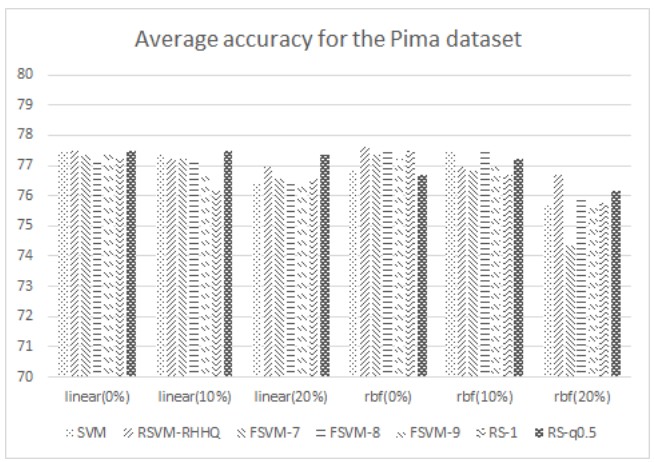

Fig. 3. Average accuracy for the Pima dataset .

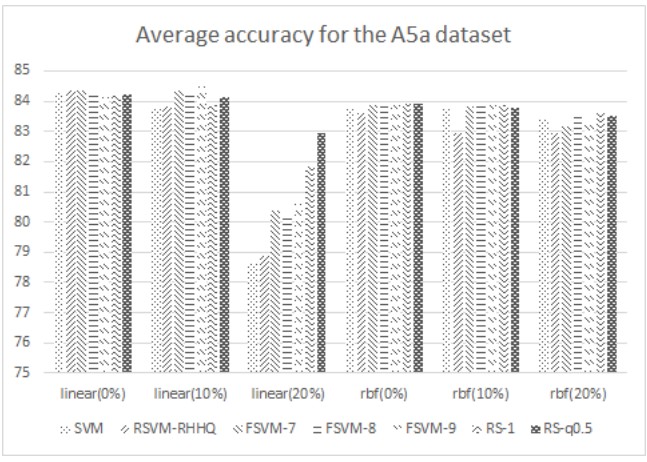

Fig. 4. Average accuracy for the A5a dataset .

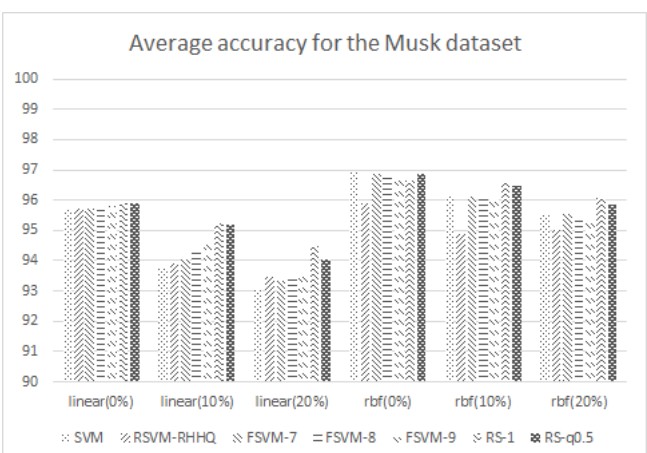

Fig. 5. Average accuracy for the Musk dataset .

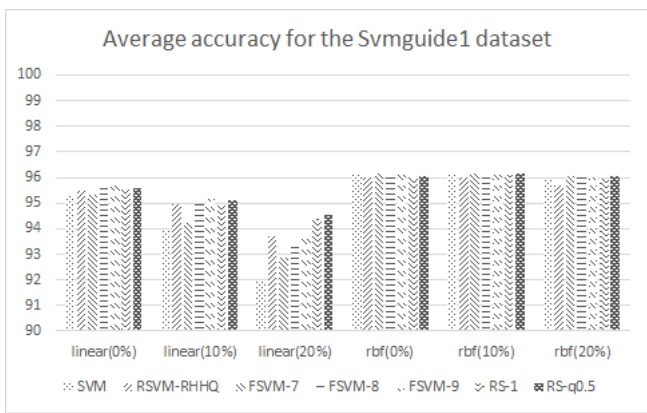

Fig. 6. Average accuracy for the Svmguide1 dataset .

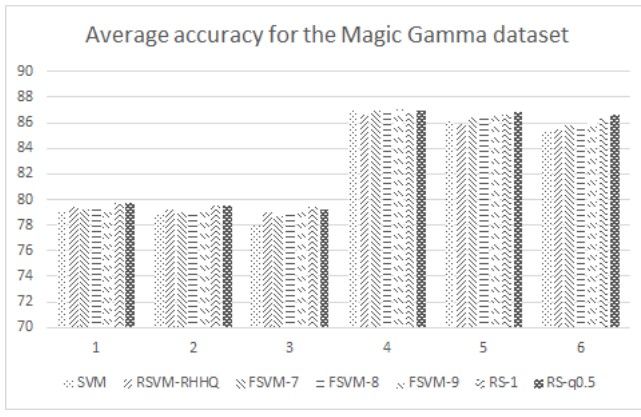

Fig. 7. Average accuracy for the Magic Gamma dataset .

its performance fluctuates in certain cases, indicating inconsistency in accuracy enhancement. In contrast, our method consistently enhances accuracy across diverse datasets, underscoring its reliability and robustness. This highlights the efficacy of our approach, especially in real-world scenarios vulnerable to interference and noise.

Furthermore, when comparing the performance of RSVM-SS-1 with RSVM-SS-q0.5, we observe that the latter demonstrates greater stability and achieves a more substantial enhancement in accuracy. This preference for using quantiles as the filtering threshold suggests that it offers a more rational approach to sample selection, thereby contributing to the overall robustness and reliability of our model.

In conclusion, our experimental findings provide compelling evidence of the effectiveness and versatility of our proposed model in enhancing the robustness and accuracy of SVM in the face of interference. These results pave the way for its application in a wide range of domains where reliable and robust classification models are essential for accurate decision-making.

### B. Model training time

Efficiency of model training is a crucial aspect in practical applications, impacting resource allocation and real-time decision-making. In this section, we investigate the training time of each model across various datasets and scenarios. Understanding the computational efficiency of each model is essential for assessing its practical viability. Fig. 9 to 15 present the training time of each model, measured in seconds. During experiments, we maintained a consistent number of iterations at 2, ensuring comparability among all models except the standard SVM. The same as Section 4.1, each dataset has six scenarios, namely, (linear, 0%), (linear, 10%), (linear, 20%), (rbf, 0%), (rbf, 10%), and (rbf, 20%).

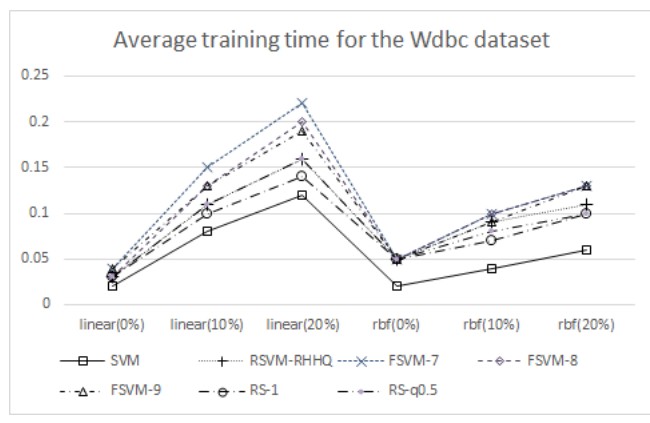

Fig. 8. Average training time for the Wdbc dataset .

Fig. 9. Average training time for the Wdbc dataset. The horizontal axis represents the combinations of different noise levels and kernel function types, where "linear" represents the linear kernel, "rbf" represents the radial basis function kernel, and the numbers represent the proportion of interference samples. The vertical axis represents the average training time of the models in seconds. Fig. 10 to 15 take the same horizontal and vertical axes.

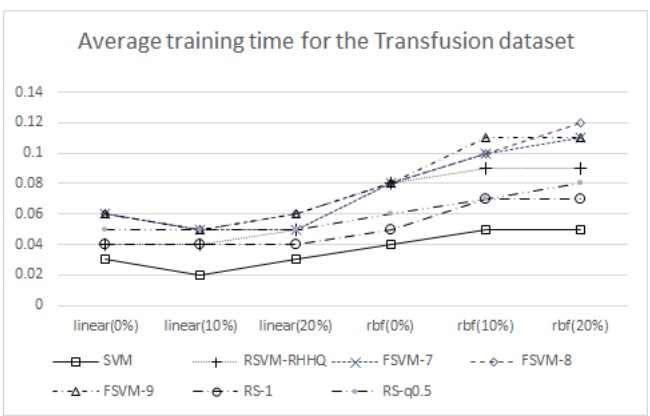

Fig. 10. Average training time for the Transfusion dataset .

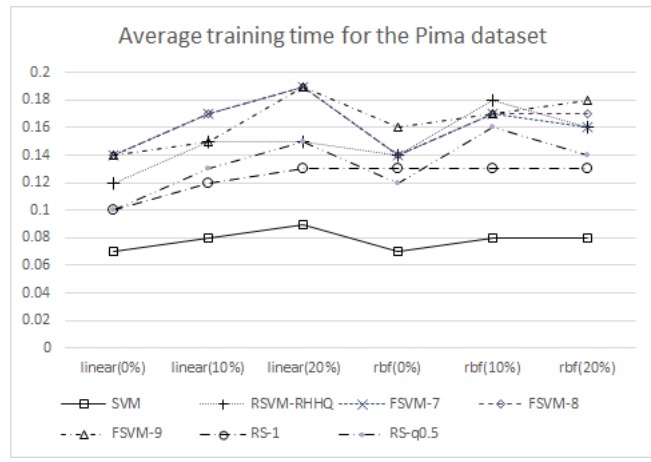

Fig. 11. Average training time for the Pima dataset .

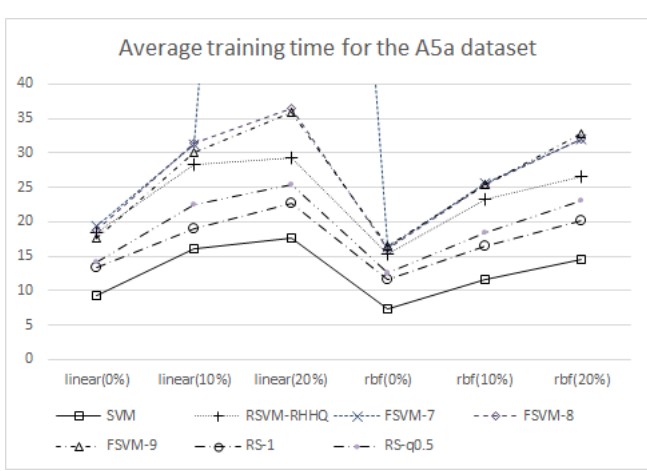

Fig. 12. Average training time for the A5a dataset .

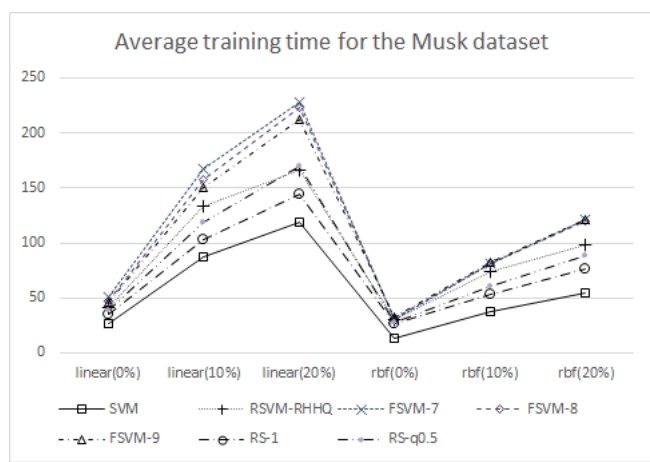

Fig. 13. Average training time for the Musk dataset .

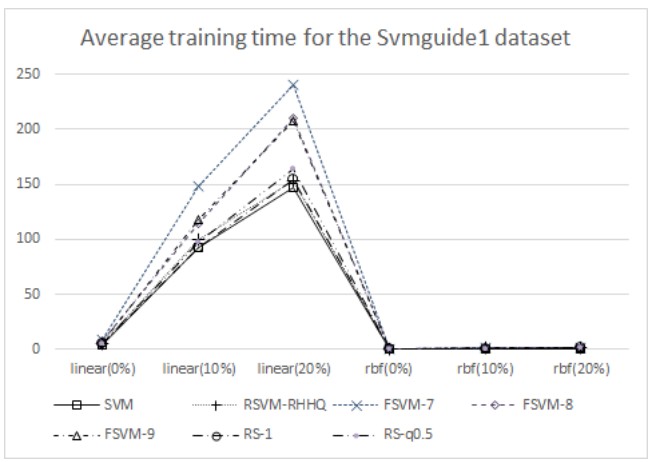

Fig. 14. Average training time for the Svmguide1 dataset .

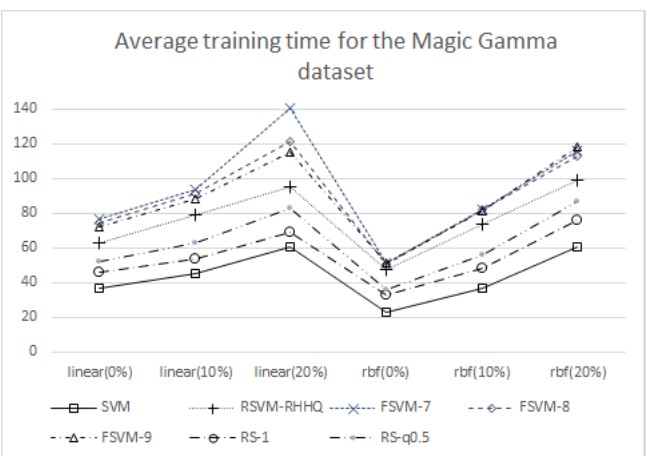

Fig. 15. Average training time for the Magic Gamma dataset .

From the above results, it is evident that RSVM-SS-1 consistently exhibits the shortest training time across all datasets, reflecting its efficiency in model convergence and parameter optimization. This efficiency is particularly notable given its competitive performance in accuracy improvement, as showcased in previous sections. While RSVM-SS-q0.5 may not match RSVM-SS-1 in terms of training time efficiency, it consistently outperforms other benchmark models, striking a balance between computational speed and predictive capability. This suggests that RSVM-SS-q0.5 offers a compelling trade-off between efficiency and accuracy, making it a practical choice for scenarios where both factors are crucial.

Moreover, the observed variations in training time among different models underscore the importance of considering computational efficiency alongside predictive performance. In real-world applications where computational resources are limited or time constraints are stringent, the choice of model can significantly impact operational efficiency and resource allocation. Therefore, a comprehensive evaluation that considers both accuracy and training time is essential for informed decision-making in model selection.

In conclusion, while RSVM-SS-1 stands out for its exceptional training time efficiency, RSVM-SS-q0.5 emerges as a strong contender, offering a pragmatic solution that balances computational speed with predictive accuracy. By integrating these findings into model selection processes, practitioners can optimize both computational resources and predictive performance, thereby enhancing the effectiveness and efficiency of machine learning applications.

## V. Conclusion

This paper has introduced a novel SVM (RSVM-SS algorithm) by proposing a new form of weights, resulting in the development of FSVM, which can be interpreted as an SVM with partial sample screening. Our experimental findings highlight the robustness of the proposed model in handling datasets prone to interference, underscoring its effectiveness in real-world scenarios characterized by noisy or complex data. Moreover, comparative analysis against other FSVM variants has revealed that our model not only demonstrates superior performance but also requires less training time, enhancing computational efficiency without compromising predictive accuracy.

Future research may explore alternative approaches for weight selection and investigate the performance of FSVM in scenarios requiring multiple iterations. Additionally, scalability and adaptability assessments across diverse datasets and application domains would enhance our understanding of FSVM's utility in practical machine learning tasks. Through continued exploration and refinement, FSVM shows promise for effectively addressing complex data challenges and facilitating decision-making across various domains.

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
