# OpenReview forum: "Robust support vector machine based on sample screening"
_IEEE.org/ICIST/2024/Conference — IEEE ICIST 2024 Conference Submission_

### Official Review · Reviewer_mUHV · 2024-08-21

[review text omitted: it was posted to a different submission]

---

### Official Review · Reviewer_LTm2 · 2024-08-22
**This article is quite fascinating and of high quality.**

**Rating:** 7
**Confidence:** 3

**Review:**

The paper titled "Robust support vector machine based on sample screening"  introduced a novel SVM (RSVM-SS algorithm) by proposing a new form of weights, resulting in the development of FSVM. Firstly, a novel robust SVM framework is introduced to counteract the effects of adversarial samples during training. Finally, the weight of samples with large loss values is dynamically set to zero to reduce the impact of outliers. This can be seen as adding sample screening to the training process, thereby reducing training time. My specific feedback is as follows: 1) Contribution 1 lack of comparison of similar methods. 2) How does adjusting sample weights affect outliers?

---

### Official Review · Reviewer_P6bR · 2024-08-23
**this work is well organized and appears potentially interesting, it can be accepted with a little modification.**

**Rating:** 7
**Confidence:** 3

**Review:**

This article, titled "Robust support vector machine based on sample screening," mainly accomplishes the development of a novel robust support vector machine (SVM) framework aimed at mitigating the impact of outliers and adversarial samples during training. The authors propose a sample weight iteration strategy, RSVM-SS (Robust SVM based on Sample Screening), which dynamically sets the weights of samples with substantial loss values to zero, effectively reducing the influence of outliers on the model. In general, this work is well organized and appears potentially interesting, it can be accepted with a little modification.
1.	Improper use of articles or long sentence structures with separators might make it difficult to follow the paper. For example, the use of ‘the’ in the paper and the presentation of subordinate clauses. Please check the full text again and modify the grammar problems.
2.	How does your paper significantly go beyond existing results?
3.	Please highlight the contributions of the paper.
4.	Please add the necessary comments for Figures.

---

### Decision · Program_Chairs · 2024-09-06

Accept (Oral)